# Association Between Congenital Gastrointestinal Malformation Outcome and Largely Asymptomatic SARS-CoV-2 Infection in Pediatric Patients—A Systematic Review

**DOI:** 10.3390/jcm14238533

**Published:** 2025-12-01

**Authors:** Iulia Stratulat-Chiriac, Elena Țarcă, Raluca Ozana Chistol, Ioana-Alina Halip, Viorel Țarcă, Cristina Furnică

**Affiliations:** 1Faculty of Medicine, Grigore T. Popa University of Medicine and Pharmacy, 700115 Iasi, Romania; chiriac.iulia@d.umfiasi.ro (I.S.-C.); alina-ioana.grajdeanu@d.umfiasi.ro (I.-A.H.); cristina.furnica@umfiasi.ro (C.F.); 2Department of Surgery II—Pediatric Surgery, Grigore T. Popa University of Medicine and Pharmacy, 700115 Iasi, Romania; 3Department of Morpho-Functional Sciences I, Faculty of Medicine, Grigore T. Popa University of Medicine and Pharmacy, 700115 Iasi, Romania; raluca-ozana.chistol@umfiasi.ro; 4Faculty of Medicine, Apollonia University, Strada Păcurari nr. 11, 700511 Iasi, Romania; viorel.tarca@univapollonia.ro

**Keywords:** gastrointestinal tract malformations, SARS-CoV-2 infection, pediatric patients

## Abstract

**Objective.** Limited evidence is available concerning the surgical outcomes of patients with congenital gastrointestinal malformations and perioperative SARS-CoV-2 infection. This study examines the scientific evidence on SARS-CoV-2 infection and congenital gastrointestinal malformations requiring surgery in children. **Material and Methods.** We performed a systematic review of studies reporting data on children with congenital gastrointestinal malformations and SARS-CoV-2 infection, published in international databases (PubMed and Embase) from pandemic inception up to August 2024. Studies not reporting data on the SARS-CoV-2 infection status on patients with congenital digestive malformation were excluded. We assessed the quality of the included studies according to the Joanna Institute (JBI) appraisal checklist, adhered to the Preferred Reporting Items for Systematic Reviews and Meta-Analysis (PRISMA) guidelines, and registered the protocol with the PROSPERO database (CRD42024550744). **Results.** From the 902 titles retrieved, eight observational studies met the inclusion criteria comprising 29 patients from countries with different socioeconomic statuses. Most patients were neonates (75%) with a median age of 3 days at diagnosis and male to female ratio of 2:1. In total, 18 (62%) presented upper gastrointestinal tract anomalies, including esophageal atresia ± tracheoesophageal fistula (*n* = 10, 34.48%), duodenal atresia (*n* = 3, 10.3%), and congenital hypertrophic pyloric stenosis (*n* = 5, 17.2%). Lower digestive tract malformations (11, 38%) included anorectal malformations (*n* = 6, 20.6%), intestinal atresia (*n* = 3, 10.3%), Hirschsprung disease (*n* = 1, 3.44%), and Meckel’s diverticulum (*n* = 1, 3.44%). Surgeries were primarily emergency or urgent procedures and only pyloromyotomy (5/5) was consistently operated minimally invasively. SARS-CoV-2 infection was identified mainly on routine screening (>95%). Of 29 patients, 85% were discharged home, and no postoperative surgical mortality and significant complications directly associated with COVID-19 were identified, although routine postoperative morbidity not linked to SARS-CoV-2 was observed. **Conclusions.** Pediatric patients with congenital gastrointestinal malformationsand perioperative SARS-CoV-2 infection typically have mild illness and favorable surgical outcomes. SARS-CoV-2 positivity alone should not delay essential surgery when infection control measures are ensured. Standardized, multicenter studies are needed to clarify perioperative risks to and inform management of this high-risk group.

## 1. Introduction

Congenital malformations are structural changes that are phenotypically and etiologically different and mainly present at birth due to disruptions of normal developmental pathways during organogenesis [1,2]. They are responsible for significant morbidity and mortality worldwide, despite advancements in antenatal screening, neonatal care, and surgical techniques, and are the fifth leading cause of death in children <5 years of age globally, according to the World Health Organization (WHO) [3]. Their prevalence varies depending on studies, region, and country income level, and almost 95% of congenital malformations are reported in low–middle income countries [4].

Congenital gastrointestinal (GI) anomalies are among the major groups of birth defects according to international surveillance data from population-based registries, with roughly two-thirds presenting as isolated cases while the remaining one-third occur in association with other congenital malformations that may significantly influence the prognosis [1,4]. For European countries, data from the European Registration of Congenital Anomalies and Twins (EUROCAT) reported a prevalence of 18.04 (16.42–19.78) per 10,000 living births in 2022 [5].

A significant proportion of patients with congenital GI anomalies are diagnosed during the neonatal period, often requiring emergency or expedited surgical intervention [6,7]. However, depending on the severity, certain anomalies may remain undetected and be diagnosed later in childhood or even in adulthood [8,9,10]. Antenatal diagnosis varies according to the specific anomaly and is influenced by prenatal care access and the quality of imaging technology.

Surgical management, whether open or minimally invasive, remains the definitive treatment for congenital gastrointestinal malformations. In many cases, delays in operative treatment lead to increased long-term morbidity and mortality with broader social and economic implications [4,11]. Conversely, selected cases that are not life-threatening, such as incidental diagnosis of Meckel’s diverticulum or short-segment Hirschsprung disease, can be safely and feasibly managed through elective surgical repair without additional adverse outcomes [6,12]. Surgical outcomes for congenital gastrointestinal anomalies vary worldwide, depending on the socio-economic status of the country. Higher incidence and worse outcomes are consistently reported for low- and middle-income countries compared with high-income countries, a gap that became more pronounced during the COVID-19 pandemic [13,14].

The recent pandemic caused by severe acute respiratory syndrome coronavirus (SARS-CoV-2) challenged and changed medical systems worldwide [15]. To sustain surgical care for patients while reducing the risks of viral transmission, protocols and guidelines were implemented for every surgical specialty. In pediatric surgical services, treatment was prioritized based on urgency. Elective procedures were initially deferred and non-surgical management was encouraged when appropriate. Additionally, minimally invasive procedures were not recommended due to concerns about aerosolization [16,17,18].

In pediatric populations, SARS-CoV-2 infection typically follows a milder clinical course and presents better prognoses than adults, although atypical clinical manifestations, such as gastrointestinal symptoms, may occur even before or in the absence of respiratory symptoms [19]. Among pediatric patients, neonates are particularly vulnerable to rapid progression to respiratory distress because of immature immune systems and the absence of vaccine protection against COVID-19 due to ineligibility for vaccination [20,21].

The impact of coronavirus disease on the surgical management and outcomes in pediatric patients with surgical digestive anomalies remains poorly defined because of the recent pandemic [22]. Emerging evidence on children diagnosed with SARS-CoV-2 infection and underlying comorbidities, including congenital anomalies, suggest an increased risk of developing severe forms of coronavirus disease. Nevertheless, current evidence on intra- and postoperative courses of patients with both congenital GI malformations and perioperative COVID-19 infection is scarce, and the effects of this association remain uncertain. This review aims to investigate the impact of perioperative SARS-CoV-2 infection on pediatric patients with congenital GI malformations in terms of surgical outcomes, including postoperative complication and mortality. Secondary objectives were to identify whether SARS-CoV-2 infection should be considered a predisposing risk factor for adverse outcomes in this selected population and to explore whether antenatal infection represents a potential etiologic risk factor for developing congenital gastrointestinal malformations. To address these objectives, we systematically reviewed the published literature including observational studies on pediatric patients with congenital GI conditions requiring surgery and SARS-CoV-2 infection.

## 2. Materials and Methods

### 2.1. Search Strategy

A systematic literature review was conducted between 15 September 2024 and 30 October 2024, examining cases of congenital GI malformations with concurrent laboratory-confirmed diagnosis of coronavirus disease. Online bibliographical databases (PubMed and Embase) were searched using relevant keywords to find articles published between January 2020 and August 2024 by using a retrospective observational approach (PICO process) as follows:Population: Pediatric patients with congenital GI malformations.Intervention: Concurrent SARS-CoV-2 infection.Comparison: None.Outcomes: Clinical outcomes (intra- and postoperative complications, surgical approach, and postoperative outcomes and mortality).

Only articles published in English were included, and no other filter was applied for the online database search.

We evaluated articles from the academic engine research using the following keywords to find relevant articles about the diagnosis and management of congenital GI malformations in children also diagnosed with SARS-CoV-2 infection: “COVID-19” OR “coronavirus” OR “Sars-CoV-2” OR “pandemic” were used in combination with “gastrointestinal duplication” OR “congenital gastrointestinal malformations” OR “congenital esophageal atresia” OR “congenital tracheoesophageal fistula” OR “esophageal duplication” OR “congenital esophageal stenosis” OR “congenital esophageal diverticulum” OR “gastric duplication” OR “congenital gastric diverticulum” OR “gastric antral web” OR “congenital gastric web” OR “microgastria” OR “pyloric atresia” OR “congenital hypertrophic pyloric stenosis” OR “duodenal atresia” OR “congenital duodenal stenosis” OR “annular pancreas” OR “congenital duodenal web” OR “duodenal duplication” OR “jejunal atresia” OR “ileal atresia” OR “malrotation” OR “congenital ileal stenosis” OR “congenital jejunal stenosis” OR “ileal duplication” OR “jejunal duplication” OR “Meckel” OR “omphalomesenteric” OR “colon atresia” OR “anorectal malformations” OR “congenital rectal stenosis” OR “recto-vestibular fistula” OR “cloacal anomaly” OR “rectal atresia” OR “Hirschsprung disease” OR “recto-perineal fistula” OR “newborn surgery and gastrointestinal” OR “neonate and surgery” OR “neonate and surgery and gastrointestinal”.

Additional article research was performed using the “related article” facility in PubMed, and references contained within relevant studies were reviewed. Their corresponding abstracts and full articles were also accessed if relevant. The systematic review protocol was developed in accordance with the Preferred Reporting Items for Systematic Reviews and Meta-Analysis (PRISMA) guidelines and registered with the PROSPERO database (no. CRD42024550744) [23].

### 2.2. Inclusion and Exclusion Criteria

Observational studies reporting data on the association between COVID-19 and congenital GI malformations in children were included in this review if they met all the following criteria: (1) full text available in English, (2) reported surgical data and/or surgical outcomes in children with congenital GI anomalies diagnosed with concurrent SARS-CoV-2 infection, (3) published since January 2020, (4) reporting cases since 1st January 2020 until August 2024. Review articles or studies not reporting original data were excluded. The pediatric population was considered under 18 years of age. Online bibliographical databases (PubMed and Embase) were searched using relevant keywords to find articles published between January 2020 and August 2024 that reported data on patients with congenital gastrointestinal malformations and coronavirus infection.

We excluded studies that (1) were duplicates, (2) did not provide information in a suitable format or did not enable separation of the data for patients with congenital GI anomalies from the data of the whole studied population, (3) had surgical/postoperative outcomes that were not retrievable from published data, (4) were not available for full-text assessment, or (5) studies not reporting data on the SARS-CoV-2 infection status of patients with congenital digestive malformations. Gray literature was screened for additional data, including academic papers, theses and dissertations, research and committee reports, conference papers, and ongoing research.

The following information was extracted: study data (author, country, year of publication, study design, and sample size), demographic data (age, gestational age, birth/current weight, and gender), comorbidities, antenatal diagnosis of congenital GI malformations for neonate and coronavirus disease for the mother, clinical data (presenting symptoms for gastrointestinal anomaly and COVID-19, reason for COVID-19 testing, pre/intra/postoperative COVID-19 status, repeated PCR test result, and COVID-19 status at discharge), surgical information (diagnosis, type of surgery and type of surgical approach, operative time, and type of anesthesia), and outcomes (length of hospital stay, neonatal intensive care unit stay, time to full feeds, intraoperative and postoperative complications, reoperation, type of discharge, re-admission, and mortality at 30 days).

Infantile hypertrophic pyloric stenosis (ICD-10 code Q40.0) was included among the congenital gastrointestinal anomalies in accordance with WHO and EUROCAT classification systems, ensuring comparability with international registry data.

### 2.3. Study Selection, Data Extraction, and Synthesis

A consecutive staged selection process was applied independently by two independent authors/researchers. The first stage involved identifying and excluding duplicates, followed by screening all titles and abstracts for potential relevance; those considered eligible were retrieved for further analysis. The same authors reviewed the full text, and if the inclusion and exclusion criteria were satisfied, data was extracted in Microsoft Excel (Microsoft Office). Disagreements were discussed with the study team and solved by consensus. Only the most recent article was included in the review for studies that reported data from the same database/hospital. No automation tools were used in the article’s selection process. For studies that did not fully meet the inclusion criteria but included data that could contribute to our analysis (e.g., case reports/retrospective studies from during the pandemic with no data on COVID-19 or surgery), one of the researchers made attempts to gain further information by emailing the authors. If there was no reply within 2 weeks, the study was excluded.

Given the nature of the available data identified in the research process, specifically the significant variation in study design and reporting data, we anticipated that meta-analysis would not be appropriate. Therefore, we performed a qualitative summary of the results from all included studies. Statistical analyses for a contingency table with small, expected frequencies were conducted using IBM SPSS Statistics, version 2.9, and used the Fisher–Freeman–Halton exact test.

### 2.4. Risk of Bias Assessment

The reviewers that identified the articles also independently assessed the quality of the included case reports, case series, and cohort studies according to the Joanna Briggs Institute (JBI) appraisal checklist for case reports, case series, and cohort studies. The JBI appraisal checklist for case series/case reports consists of eight questions for case reports and ten questions for case series related to the absence or existence of various reported items. For cohort studies, the JBI checklist consists of 12 questions. We considered articles as high-quality if reported items addressed more than two-thirds of the JBI checklist and moderate-quality if more than one-third of items were addressed. Articles with reported items addressing less than one-third of the JBI checklist were deemed low-quality.

## 3. Results

### 3.1. Study Retrieval Strategy

Our research strategy identified 902 studies. After screening for duplicates, 273 entries were eliminated, and 629 progressed to abstract and title screening. Of these, 595 proved to be irrelevant, and for the remaining 34, the full text was retrieved and subjected to appraisal. Of these, eight studies contained data that could contribute to our analysis but had incomplete information to fulfill the inclusion criteria on either COVID-19 information or surgery (management or outcomes). Hence, we emailed the corresponding authors inquiring about additional data and waited two weeks for a reply. We received feedback from four and nothing for the remaining authors, and data provided did not alter the exclusion decision. Six of the remaining twenty-six articles met the inclusion criteria after a full-text assessment. In addition, two eligible articles were retrieved following manual reference screening and the “related article” facility in PubMed for the included studies, resulting in a total of eight studies included in the analysis. The screening process is summarized in the PRISMA flow diagram—Figure 1.

### 3.2. Study Characteristics

The list of studies that included surgical outcomes are displayed in Table 1. They were all published in international journals during the pandemic, with data collection between March 2020 and July 2021. We classified studies according to their design as cohort studies, retrospective (*n* = 2) and prospective (*n* = 1) studies, case reports (*n* = 2), and case series (*n* = 2). For one publication that included a cohort of patients (25), study design, either prospective or retrospective, was not acknowledged.

All cohort studies were single-center experiences. The total sample size reported in the studies including COVID-19 was 194 patients, but only 29 (15%) were diagnosed with congenital GI anomalies and were included in the analysis. Of these, 21 (72%) were from lower-middle-income countries (Bangladesh and India) and were diagnosed with EA-TEF, duodenal atresia (DA), intestinal atresia, anorectal malformation (ARM), Hirschsprung disease (HD), or congenital hypertrophic pyloric stenosis (CHPS). Although all publications reported information on demographics, the number of studies reporting the complications, surgical timing/procedure, and postoperative outcomes varied markedly. Of the eight studies included, all were deemed to be of moderate/high methodological quality.

### 3.3. Patients’ Characteristics

Age at presentation ranged from 1 day to 1.25 years (median = 3 days), with neonates accounting for 75% of the population study. Data on antenatal diagnosis of congenital malformation, SARS-CoV-2 infection, or maternal vaccination was not reported for patients included in the study. Postnatal data such as birth weight, gestational age, and age at surgery were scarcely provided, even in case reports. Gender was reported in case series and case reports, but data was missing in cohort studies. Therefore, for the reported data, the male-to-female ratio was 2:1, and neonates were born at term gestation with a median weight of 2.500 g (IQR 2400-3000).

Regarding patient characteristics in the overall cohort, 18 (62%) had upper GI tract congenital anomalies, while 11 (38%) had lower GI malformations. Surgical details are presented in Table 2.

EA ± TEF was the most diagnosed upper tract GI anomaly (*n* = 10, 34%), followed by ARM (*n* = 6, 20%) and CHPS (*n* = 5, 17%). One patient with a colostomy for Hirschsprung disease was admitted with a prolapsing stoma and underwent emergency definitive surgical treatment (pull-through) using a minimally invasive approach.

Open procedure (laparotomy/thoracotomy) was performed for EA-TEF, DA, Meckel diverticulum, and anorectal malformations, with one case of duodenal atresia that started minimally invasive and converted to open approach due to complex intraoperative surgical findings. A minimally invasive technique was successfully performed for the majority of patients diagnosed with CHPS (4/5) and one patient with Hirschsprung disease a prolapsed colostomy. Shah et al. [31] did not report procedure-specific data for the included congenital gastrointestinal malformations. The study noted a preference for open emergency surgeries and reduced minimally invasive procedures during the pandemic.

Surgical invasiveness was determined according to the operative approach (open/minimally invasive/minimally invasive converted to open) and surgical technique (presence of anastomosis or stoma). Procedures performed through minimally invasive techniques or involving limited local repair (such as pyloromyotomy) were classified as low invasiveness. In contrast, open procedures incorporating anastomosis were categorized as high invasiveness, whereas operations performed minimally invasive or transanally but involving gastrointestinal anastomosis were designated as moderate. Out of the 29 patients analyzed, surgical information was available for 13 (44.8%). Among these, four patients (30.8%) underwent procedures classified as low invasiveness, three (23.1%) as moderate, and six (46.1%) as highly invasive.

Operative time was reported in two publications only for minimally invasive procedures for pyloromyotomy as follows: <40 min [27] and 73 min [25]. A time of 219 min was necessary for resection pull-through laparoscopy in a patient with a prolapsed colostomy and Hirschsprung disease [25]. All surgical interventions were performed under general anesthesia, and the length of postoperative stay varied between 1 and 60 days with a median of 4 days.

Overall, for the reported data, the postoperative outcomes were favorable, with 84% of patients being discharged home and one neonate transferred to a coronavirus-designated hospital. Complications likely associated with COVID-19 were observed in two patients (15.4%), both of whom had undergone highly invasive interventions. Additionally, one patient (7.7%) from the same category experienced a surgery-related complication. No complications were reported in the low- or moderate-invasiveness groups.

Respiratory complications were documented in two studies [28,29], each affecting one patient; however, no definitive association with COVID-19 infection was identified. An intraoperative anesthetic complication during duodenal atresia repair was noted when the patient developed hypercarbia after intubation and remained intubated postoperatively for 24 h; however, this was anticipated due to comorbidities [29]. Moreno-Duarte et al. reported high ventilatory requirements during laparoscopic duodenal atresia that was converted to open due to the complexity of the congenital gastrointestinal anomaly. The authors associated the anesthetic complication with HEPA (High-Efficiency Particulate Air) filters usage during surgery. Another neonate from Bangladesh with neonatal sepsis died at 4 days of life before receiving surgery for high anorectal malformation. This case was retained in the overall cohort to reflect the full clinical population but registered as a preoperative mortality, as the event was unrelated to either the surgical procedure or COVID-19 infection. As shown in Table 3, all complications occurred in the high-invasiveness group; however, Fisher–Freeman–Halton exact testing revealed no statistically significant association between surgical invasiveness and either COVID-19-related (*p* = 0.30) or surgery-related (*p* = 0.47) complications.

Shah et al., in a cohort of 31 patients with congenital diseases (15 with GI anomalies) and COVID-19 infection, reported good outcomes and no adverse events resulting from coronavirus disease for 24 patients. However, seven neonates were lost (death/lost at follow-up) with no cause reported. Of 31 surgical patients with COVID-19 infection, 19 were neonates and, in addition to GI malformations, there were other diagnoses such as omphalocele major (*n* = 1), neonatal necrotizing enterocolitis (*n* = 2), cystic lung lesion (*n* = 1), and lumbosacral myelomeningocele (*n* = 1) that could lead to a poor outcome.

### 3.4. COVID-19 Characteristics

SARS-CoV-2 was detected from a nasopharyngeal specimen using mainly reverse transcription-polymerase chain reaction assay (RT-PCR, 90%) or, less commonly, the Biofire Filmarray Respiratory Panel Test [28,29]. Table 3 provides data on coronavirus disease features for included patients. SARS-CoV-2 testing was mainly performed as per diagnostic protocol prior to surgical intervention (96.55%) and not because of symptoms. One neonate from a low–middle income country presented with early onset of neonatal sepsis and imperforate anus, additionally diagnosed with SARS-CoV-2 infection on admission, and succumbed on day of life 3.

Patients were rarely tested postoperatively unless the test was not performed on admission and there was either a change in the patient’s clinical status or other medical evidence, such as reported by Bindi et al. for the neonate whose referral pediatrician tested positive for coronavirus disease [24]. For the neonatal cohort, maternal testing was negative for 17%, positive at five days post-delivery for 4%, and no data was reported for 70%. For two neonates born in LMICs, their mothers were not tested for coronavirus disease, likely due to reduced financial resources.

Only one study [26] reported data on delaying surgery until COVID-19 tests were negative, and this approach had no surgical intra- or postoperative complications. This approach was possible for three out of four patients who were medically managed in the NICU while expecting surgery for EA-TEF (*n* = 2) and duodenal atresia with pyloric web. For one neonate with ARM requiring stoma formation, surgery was urgently performed while waiting for coronavirus RT-PCR results that returned positive (Table 4).

For emergent or life-threatening cases when the COVID-19 test result was not available, patients were treated as patients under investigation and taken to the operating room prior to the test result. Appropriate coronavirus protocols were applied for the safety of both the patient and the surgical team.

All children diagnosed with congenital GI malformation and concurrent SARS-CoV-2 infection among all included studies in our systematic review were not vaccinated against SARS-CoV-2. Antenatal SARS-CoV-2 testing or maternal coronavirus disease was not reported in the studies included (Table 4).

The collective evidence from multiple tertiary surgical centers demonstrates that perioperative care for surgical patients with congenital gastrointestinal malformations during the coronavirus pandemic was feasible because systematic protective measures were applied. Core strategies for patients and personnel protections are summarized in Table 5.

## 4. Discussions

This study provides an overview of current evidence on the association between SARS-CoV-2 infection and congenital gastrointestinal malformations in children based on case series, case reports, and cohort studies published since the onset of the COVID-19 pandemic. In our series of patients, the clinical course was largely mild in severity with infrequent perioperative complications and no mortality directly linked with COVID-19 infection. The findings align with emerging reports that identified low post-operative complication rates and an absence of mortality directly attributable to SARS-CoV-2 infection, suggesting that perioperative SARS-CoV-2 infection is not a primary determinant of outcomes, even among neonates, and surgical outcomes depend more on the type congenital anomaly and associated comorbidities [32,33].

All congenital gastrointestinal anomalies included in this review were coded according to the International Classification of Diseases, 10th Revision (ICD-10, Q39–Q45) to ensure consistency with international standards. Although infantile hypertrophic pyloric stenosis (IHPS) usually becomes symptomatic after birth, it is designated as a congenital malformation in ICD-10 (Q40.0) and in the EUROCAT registry system. This classification reflects a presumed prenatal developmental origin and enables comparison with global congenital anomaly data [34,35].

Concerns have been raised worldwide related to the teratogenic effect of antenatal SARS-CoV-2 infection or maternal COVID-19 vaccination [36,37,38]. However, recent population-based cohort studies encompassing large numbers of liveborn infants since the onset of the pandemic have reported no significant increase in the incidence of congenital GIT malformation development after maternal coronavirus infection or vaccination during pregnancy [39,40,41,42]. In studies included in this review, there is no reported detailed maternal history, particularly on antenatal SARS-CoV-2 infection or maternal vaccination. Consequently, findings of this review cannot substantiate the teratogenic hypothesis. The fact that all patients were diagnosed with a congenital GI anomaly within the neonatal period reflects the usual embryologic cause rather than COVID-19 exposure.

In children, SARS-CoV-2 infection generally presents milder clinical forms, faster recovery, and more favorable prognosis than adults. The landmark COVIDSurg/GlobalSurg international cohort (primarily adult patients) reported a 23.8% 30-day mortality and a pulmonary complication rate of 51.2% among perioperatively infected patients, and mortality exceeded 38% in patients that developed pulmonary complications [41]. In children, evidence from emerging reports remains variable. While certain studies reported that pre-existing conditions such as congenital anomalies, including GI malformations, may increase the risk of developing severe forms of disease [20,43,44], other investigators found no significant differences in surgical outcomes or pulmonary complications related to SARS-CoV-2 infection [21,45,46,47,48]. Among the cases analyzed in this review, additional diagnosis of perioperative SARS-CoV-2 infection was established mainly on screening and not because of symptoms or complications attributable to SARS-CoV-2 infection, and it did not alter the surgical diagnosis or outcomes. Similar findings were reported by Nepegodiev et al. and GlobalSurg Collaboratives [49]. Reassuringly, no death occurred intra and postoperatively, but the mortality rate for patients included in this review was 3.5%. (*n* = 1). This was reported from Bangladesh in a neonate admitted with an imperforate anus and neonatal sepsis who succumbed the day following admission prior to surgical input. Farooq et al., in a study from Bangladesh during the coronavirus pandemic, found that neonatal surgical admission remained constant but neonatal mortality increased, with ARM and intestinal obstruction as the leading causes of death, likely worsened by a decline in essential care during the pandemic [14,26,30,50,51,52]. All postoperative and COVID-19-related complications occurred in patients who underwent highly invasive procedures, suggesting that surgical complexity may increase the risk of adverse outcomes. The absence of complications in less invasive procedures likely corresponds to reduced perioperative stress and operative trauma. Given the limited sample size and incomplete surgical data, these observations should be interpreted cautiously. No statistically significant relationship was identified between the degree of surgical invasiveness and occurrence of complications (surgery- or COVID-19-related) in our limited cohort of patients. Given the small sample size, the statistical power of analysis is reduced, and results should be interpreted cautiously. However, these observations align with the current pediatric surgical literature, which suggests that perioperative COVID-19 infection does not necessarily worsen the operative outcomes, which may be influenced by more complex surgical procedures [53].

Timing surgical intervention is essential in children with congenital gastrointestinal malformations, as delays are associated with increased mortality and long-term morbidity rates. During the coronavirus pandemic, although healthcare systems experienced substantial disruption, emergency surgical treatment for patients with congenital GI malformations continued to be performed without substantial variation. This observed stability indicates that the prevalence of congenital gastrointestinal malformation was not significantly influenced by the coronavirus pandemic [36,42,54]. Patients included in this review predominantly underwent emergency surgeries, with only a small subset—two cases with esophageal atresia and one with duodenal atresia—classified as semi-elective. The maintenance of timely surgical treatment was achieved through deliberate institutional resource optimization responding accordingly to the pandemic requirements.

Surgical approach during the pandemic was influenced not only by patient-related factors such as SARS-CoV-2 infection or vaccination status but also by hospital policies and pandemic control measures, which varied according to institutional policies determined by resource availability and infection-control requirements [54,55,56]. Although minimally invasive surgery was initially not recommended because of concerns about viral transmission [13,18], surgical societies released interim guidance to help surgeons provide optimal patient care, maintain staff safety, and help contain the pandemic. Nonetheless, surgical approach was based on individual discernment and expertise and available resources [16]. In the cohort of 29 patients included in this review, a reduced use of minimally invasive techniques (25%), especially for the upper tract anomalies, was observed, results consistent with the recommended protocols. However, when feasible and when the benefits outweigh the risks, minimally invasive surgery should remain the preferred treatment option for congenital gastrointestinal malformations.

Most patients in this review were from low- and middle-income countries, where limited neonatal resources and pandemic-related constraints may have contributed to poorer outcomes. The single reported death, occurring in a neonate from Bangladesh with an imperforate anus and sepsis, reflects the challenges of delayed presentation and restricted critical care access. These findings underscore persistent global inequities in pediatric surgical care and the urgent need to strengthen neonatal surgical capacity in resource-limited settings.

Long-term outcomes in children with congenital malformations are predictable, but the consequences of infection with coronavirus disease, especially in the neonatal period, are yet to be determined. This period is also more vulnerable in terms of growth and development as infections contracted during this time might disrupt normal development. Emerging reports from COVID-19-positive pregnancies and congenital GI-operated malformations over a 12-month period identified growth and developmental delays and more common gastrointestinal health issues [57], but long-term follow-up is necessary to identify the impact on children’s growth and development.

There are several limitations to our study. Given the recent coronavirus pandemic and scarce medical reports on the specific association between congenital GI malformations and SARS-CoV-2 infection, the current evidence consists of retrospective studies, with inherent methodological constraints such as data completeness and reporting variability. The small sample size (*n* = 29) restricts the statistical power and limits the generalizability of the findings. Consequently, results should be interpreted with caution. Considerable heterogeneity was also observed across included studies, reflecting differences in patient age, anomaly type, healthcare settings (high- vs. low-income countries), and study design. To address this, we used narrative synthesis of data to ensure transparent interpretation of evidence. While this variability reduces the strength of the conclusion, it also reflects the real-world diversity in practice.

The PICOS framework could not be applied because of different study designs each contributing various levels of evidence (retrospective/prospective/case series and case reports) and hence, the PICO structure was adopted to report all types of clinical evidence, maintaining the scope of this review, but our conclusions should be interpreted considering these limitations.

Overall, pediatric patients with congenital gastrointestinal malformations and COVID-19 demonstrated favorable short-term outcomes. However, the current evidence is limited by small sample sizes and inconsistent reporting of key prognostic variables, including consistent documentation of anomaly type and disease severity. Standardized reporting is needed to improve future epidemiological and outcome analyses. Another limitation is the absence of detailed data on associated congenital anomalies in most reviewed or comparable reports, which prevented further analysis of their potential impact on outcomes. Future studies should use standardized reporting to include associated anomalies, which are known to influence both the surgical outcomes and overall prognosis. As a result, potential interactions between congenital gastrointestinal malformations and other congenital abnormalities could not be evaluated.

These findings should be interpreted with caution, as most of the included patients were asymptomatic or mildly symptomatic at the time of SARS-CoV-2 detection, limiting generalizability to more severe cases. The lack of non-COVID-19 comparison groups further restrict conclusions to descriptive observations rather than assessments of the infection’s independent impact on postoperative risk. Although no complications were attributed to SARS-CoV-2, one routine surgery-related event unrelated to infection was reported, and inconsistent reporting hindered evaluation of minor morbidity. These limitations highlight the need for prospective studies across a wider spectrum of disease severity.

The coronavirus pandemic progressed worldwide on a variable scale, and the healthcare response in terms of surgical treatment standards and pandemic control was dependent on many factors, including the socio-economic status of the country [54]. Data collected from high- and low-middle income countries on COVID-19 patients has been essential to identify risks associated with SARS-CoV-2 infection in particular populations (such as neonates with congenital GI malformations requiring corrective surgery) and to inform the surgical community on updated practices. Including data from diverse care systems in high- and low-resource regions would improve generalizability, providing more representative results. However, multicenter studies employing standardized data collection and analysis are needed.

## 5. Conclusions

In summary, this case series indicates that SARS-CoV-2 infection does not influence the perioperative outcomes in pediatric patients with congenital GI malformations, though reported evidence remained limited. The clinical courses were largely mild, postoperative recovery was uneventful, and no morbidity or mortality was attributable to COVID-19. These findings support proceeding with timely surgical intervention when clinically indicated, assuming adequate infection control protocols. However, the limited and heterogeneous data underscore the need for collaborative, methodologically robust studies with comprehensive follow-up to clarify outcomes and support evidence-based clinical recommendations. In addition, because appropriate non-COVID-19 comparator groups were largely absent in the available studies, this review can describe the observed outcomes but cannot determine the independent effect of SARS-CoV-2 infection on postoperative risk. Therefore, the conclusions should be interpreted as descriptive observations rather than a definitive assessment of SARS-CoV-2-associated risk.

## Figures and Tables

**Figure 1 jcm-14-08533-f001:**
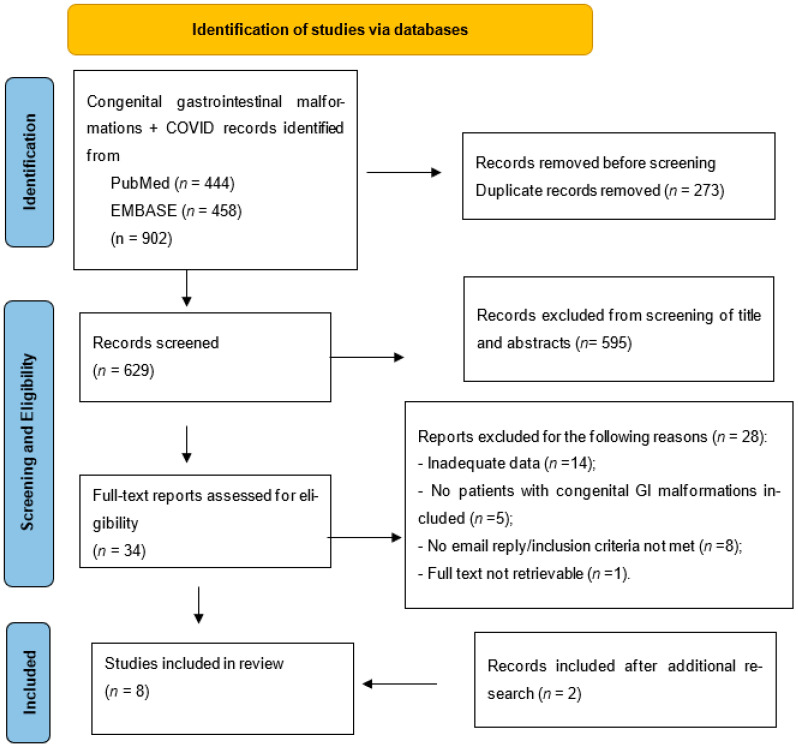
PRISMA flow diagram—study selection for review.

**Table 1 jcm-14-08533-t001:** List of included studies with study characteristics and patient outcomes.

Authors/Reference	Journal	Year Published/ Analyzed Period	Country	Study Design	No. of Patients	Surgical Diagnosis	Outcomes
Bindi et al. [24]	Journal of Pediatric Surgery Case Reports	2020	Italy	Case report	1	Meckel’s diverticulum	Discharged home
Haddadin et al. [25]	The American Surgeon	2022 20 July–21 June	USA	Prospective	2	CHPS HD	Discharged home
Kadian et al. [26]	African Journal of Pediatric Surgery	2022	India	Case series	4	EA-TEF (*n* = 2) Low ARM DA with pyloric web	Discharged home
Mehl et al. [27]	Journal of Pediatric Surgery	2021 April–20 August	USA	Retrospective	3	CHPS (*n* = 3)	Discharged home
Moreno-Duarte et al. [28]	Pediatric Anesthesia	2021	USA	Case report	1	DA	Discharged home
Saynhalath et al. [29]	Pediatric Anesthesiology	2021 April–20 September	USA	Retrospective	1	DA	No data
Saha et al. [30]	The Pediatric Infectious Disease Journal	2020 March–20 July	Bangladesh	Case series	2	ARM	1 * transfer to a COVID-19-designated hospital postoperatively 1 * death prior to surgical input (early neonatal sepsis)
Shah et al. [31]	J Indian Assoc Pediatr Surg	2022 20 March–21 July	India	Cohort	15	TEF (*n* = 8) Intestinal atresia (*n* = 3) High ARM (*n* = 3) CHPS (*n* = 1)	Partially reported: 24 patients recovered; 7 died or lost to follow-up (diagnoses unspecified) *

Abbreviations: Congenital hypertrophic pyloric stenosis (CHPS), Anorectal malformation (ARM), Esophageal atresia with tracheoesophageal fistula (EA-TEF), Duodenal atresia (DA). * No complications related to coronavirus disease reported for patients included in the study.

**Table 2 jcm-14-08533-t002:** Pediatric patients with congenital GI malformations and COVID-19 stratified by diagnosis and surgical details.

Surgical Diagnosis	No. of Patients (%)	Surgical Approach	Surgical Procedure	Surgical Invasiveness
ARM	3 (10.3%)	No data	No data [31]	NA
	1 (3.44%)	NA	NA [30]	NA
	2 (6.89%)	Transanally	Anoplasty (V-Y)	Moderate
CHPS	4 (13.79%)	Minimally invasive	Pyloromyotomy	Low
	1 (3.44%)	No data	No data [31]	NA
DA	2 (6.89%)	Open	Duodenoduodenal (+pylorojejunal anastomosis)	High
	1 (3.44%)	Minimally invasive −>converted to open	Duodenoduodenal anastomosis	High
HD	1 (3.44%)	Minimally invasive	Resection pull-through	Moderate
Intestinal atresia	3 (10.3%)	No data	No data [31]	NA
Meckel’s diverticulum	1 (3.44%)	Open	Diverticulum resection with ileo-ileal end-to-end anastomosis	High
EA ± TEF	2 (6.89%)	Open	End-to-end esophageal anastomosis± fistula closure	High
	8 (27.55%)	No data	No data [31]	NA
Total	29			

**Table 3 jcm-14-08533-t003:** Relationship between surgical invasiveness, COVID-19-related complications, and surgery-related complications.

Surgical Invasiveness	COVID-19-Related Complications	Surgery-Related Complications	Total (*n*)	% of Total Cohort
Low	0	0	4	30.8%
Moderate	0	0	3	23.01%
High	2 (15.4%)	1 (7.7%)	6	46.01%
Total	2	1	13	
	Fisher–Freeman–Halton exact *p* = 0.3	Fisher–Freeman–Halton exact *p* = 0.47		
No data/No surgery	-	-	16	

Data are presented as number of cases (percentage of total). Statistical analysis performed using the Fisher–Freeman–Halton exact test with a two-tailed significance level <0.05.

**Table 4 jcm-14-08533-t004:** Pediatric patients with congenital GI malformations diagnosed with SARS-CoV-2 infection stratified by age, COVID-19-related data for patients and mothers, surgical diagnosis, and complications (*n* = 29).

Authors	No. of Patients	Age	Clinical Status	Reason for COVID-19 Testing	Type of COVID-19 Testing	COVID-19 Test Result			Diagnosis	Complications		Mother’s COVID-19 Status
						Preop	Intraop	Postop		COVID-19	Surgical	
Bindi et al. [24]	1	3 days	No symptoms	Symptomatic referral pediatrician	No data	Not tested	Not tested	Positive	Meckel’s diverticulum	No	Yes	Negative
Haddadin et al. [25]	2	6 weeks	No symptoms	Protocol	RT-PCR	Positive	Positive	Not tested	CHPS	No	No	No data
		1.25 years	No symptoms	Protocol	RT-PCR	Positive	Positive	Not tested	HD	No	No	No data
Kadian et al. [26]	4	2 days	No symptoms	Protocol	RT-PCR	Positive	Positive	Neg POD 5	ARM	No	No	Negative
		3 days	No symptoms	Protocol	RT-PCR	Positive on admission/Neg at DOL 3 and 4	Neg	Not tested	EA-TEF	No	No	No data
		1 day	No symptoms	Protocol	No data	Positive at DOL 7 + 8/Neg at DOL 13	Neg	Not tested	DA with pyloric web	No	No	Positive after delivery (day 5)
		5 days	No symptoms	Protocol	RT-PCR	Positive on admission/Neg at DOL 4 and 6	Neg	Not tested	EA-TEF	No	No	Negative
Mehl et al. [27]	3	4–6 weeks	No data	Protocol	RT-PCR	Positive	Positive	No data	CHPS	No	No	No data
Moreno-Duarte et al. [28]	1	4 days	No symptoms	Protocol	Biofire Panel	Pending	Positive	No data	DA	Yes	No	Negative
Saynhalath et al. [29]	1	4 days	No data	Protocol	RT-PCR/Biofire Panel//Rapid test	Positive	Positive	No data	DA	Yes	No	No data
Senjuti et al. [30]	2	2 days	Early onset of neonatal sepsis	Protocol	RT-PCR	Positive	NA	NA	ARM (imperforate anus)	Succumbed at DOL 3 (neonatal sepsis)	N/A	Not tested
		1 day	No symptoms	Protocol	RT-PCR	Not tested	Not tested	Positive (POD2)	ARM (Ano cutaneous fistula)	No	No	Not tested
Shah et al. [31]	15	Neonates	No data	Protocol	RT-PCR	Positive/pending *	Positive/ pending *	No data	EA-TEF (*n* = 8) Intestinal atresia (*n* = 3) ARM (*n* = 3) CHPS (*n* = 1)	No	No ***	No data

* [patients with ARM requiring stoma were taken to theater without waiting for the COVID-19 test result; for other surgical emergencies, patients waited for test result prior to surgery]; *** [24/31 positive patients had a good surgical outcome and no adverse sequela of COVID-19, but 7 neonates lost—cause provided -> lost at follow up/death]; Abbreviations: not applicable—NA, positive—pos, negative—neg, Biofire Respiratory Panel 2019—Biofire panel, pre/intra/postop—pre/intra/postoperatively.

**Table 5 jcm-14-08533-t005:** COVID-19 protective strategies implemented as reported in articles.

Reference No	Author	COVID-19 Protective Strategies
[24]	Bindi et al.	Patients’ isolation; staff swab-tested; emphasize pre-op screening to limit exposure
[25]	Haddadin et al.	Used smoke evacuation/filtration; found no SARS-CoV-2 in aerosols but kept strict containment practices
[26]	Kadian et al.	Dedicated COVID-19-neonates intensive care and operating room; PPE for all staff; defer non-urgent cases until negative test; urgent cases under full COVID protocol
[27]	Mehl et al.	Institutional COVID perioperative pathway; testing and triage before surgery; safe workflows to continue urgent surgeries
[28]	Moreno-Duarte et al.	Airborne PPE (N95, gown, goggles, gloves); minimize staff during intubation/extubation; HEPA filters in circuit
[30]	Saha et al.	Routine SARS-CoV-2 testing for surgical patients; isolation and referral to COVID-19-dedicated hospitals
[29]	Saynhalath et al.	Highlighted increased peri anesthetic respiratory complications; reinforced airborne-level PPE and airway planning
[31]	Shah et al.	Dedicated COVID OR/team; full PPE; minimize staff during intubation/extubation; smoke evacuation; RT-PCR before elective cases

## Data Availability

No new data were created or analyzed in this study.

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
