# Peer review of "Association Between Congenital Gastrointestinal Malformation Outcome and Largely Asymptomatic SARS-CoV-2 Infection in Pediatric Patients—A Systematic Review"

_jcm, 2025, doi:10.3390/jcm14238533_

Round 1
Reviewer 1 Report
Comments and Suggestions for Authors
This review investigated the outcomes of congenital gastrointestinal malformation (CGM) cases when they contracted COVID-19. The paper's structure itself is problematic and cannot be considered a systematic review, leaving readers with no conclusions to draw. Therefore, this reviewer recommends reconsidering the paper from the structural stage. The issues are listed below.
- CHPS is included among the target diseases (Tables 2, 3). CHPS is acquired, not congenital.
- The target disease group includes both conditions that frequently co-occur with other congenital abnormalities (CA) and those that do not. The presence or absence of CA is not stated in the results. Furthermore, this is not discussed in the discussion section. For example, while CA is rare in Meckel's diverticulum, EA-TEF frequently involves VATER association, and DA often involves chromosomal abnormalities or congenital heart disease. The presence or absence of CA is expected to alter outcomes if COVID-19 is contracted significantly; therefore, CA should naturally be documented and discussed.
- The descriptions from L-305 to L-311 are presumed to pertain to Reference 25, given the case numbers. Table 1 lists "no data" for "15 with GI anomalies." However, L-306 to L-307 state "good outcomes and no adverse events resulting from coronavirus disease for 24 patients," revealing a discrepancy between the text and Table 1.
4. The severity of each case should inherently be considered. For example, EA-TEF survival rates have traditionally been predicted using the Waterstone classification. In COVID-19 cases, more severe cases tend to have poorer prognoses. Therefore, severity should be included as an investigative item when documented in the cited literature. However, depending on the documentation method in the cited literature, severity may not be discernible.
Author Response
Reviewer 1 – comments and responses
Comments - This review investigated the outcomes of congenital gastrointestinal malformation (CGM) cases when they contracted COVID-19. The paper's structure itself is problematic and cannot be considered a systematic review, leaving readers with no conclusions to draw. Therefore, this reviewer recommends reconsidering the paper from the structural stage. The issues are listed below.
Comment 1 – HPS is included among the target diseases (Tables 2, 3). CHPS is acquired, not congenital.
Response 1 –
We acnowledge the observation regarding the etiology of pyloric stenosis. According to international classification systems, International Classification of Diseases, 10th Revision (ICD-10, code Q40.0) and EUROCAT (European Surveillance of Congenital Anomalies), pyloric stenosis is coded as congenital malformation of the stomach and duodenum. This classification acknowledges that the predisposing developmental process begins before birth, even though clinical signs typically appear in the early neonatal period and was maintained to preserve consistency and comparability with existing epidemiological datasets and registries.
Accordingly, we underscored this in Material and Methods (page 4, line 177-179) and in the Discussion section. (page 10, line 371 – 377)
Comment 2 - The target disease group includes both conditions that frequently co-occur with other congenital abnormalities (CA) and those that do not. The presence or absence of CA is not stated in the results. Furthermore, this is not discussed in the discussion section. For example, while CA is rare in Meckel's diverticulum, EA-TEF frequently involves VATER association, and DA often involves chromosomal abnormalities or congenital heart disease. The presence or absence of CA is expected to alter outcomes if COVID-19 is contracted significantly; therefore, CA should naturally be documented and discussed.
Response 2 -
We appreciate the reviewer for underscoring the importance of reporting associated congenital anomalies when analysing data on congenital gastrointestinal malformations, especially when reporting results on clinical outcome. It is well described in the literature that coexisting anomalies—such as cardiac defects or chromosomal abnormalities in duodenal atresia, and VACTERL associations in esophageal atresia—can significantly affect outcomes.
Among the included studies, only one, [24], identified a case of congenital gastrointestinal malformation (Meckel’s diverticulum) clearly stating the absence of additonal congenital malformations; the remaining articles did not report specific data on congenital anomaly status. Hence, detailed analysis was not possible.
Accordingly, we have revised the limitation section. (page 12, line 469 - 476)
Comment 3 – The descriptions from L-305 to L-311 are presumed to pertain to Reference 25, given the case numbers. Table 1 lists "no data" for "15 with GI anomalies." However, L-306 to L-307 state "good outcomes and no adverse events resulting from coronavirus disease for 24 patients," revealing a discrepancy between the text and Table 1.
Response 3 –
We thank the reviere for highlighting this discrepancy between the text and the data on table 1. Revising data analysed, the authors reported data on 31 pediatric surgical patients with COVID-19 infection and associatede anomalies, including 15 patients with gastrointestinal anomalies. The text reports “good outcomes and no adverse events for 24 patients,” while noting that seven neonates died or were lost to follow-up without specifying their underlying diagnoses without specific details for the subgroup with gastrointestinal anomalies.
We have revised results from Table 1 (data on Shah et al. [25]) specifying that outcome information was reported for the full cohort and not separately for patients with gastrointestinal malformations. (page 6/7, Table 1).
Comment 4 – The severity of each case should inherently be considered. For example, EA-TEF survival rates have traditionally been predicted using the Waterstone classification. In COVID-19 cases, more severe cases tend to have poorer prognoses. Therefore, severity should be included as an investigative item when documented in the cited literature. However, depending on the documentation method in the cited literature, severity may not be discernible.
Response 4 -
We appreciated the reviwer’s suggestion. Consistent with emerging literature, severity of congenital gastrointestinal malformations affect both the surgical outcomes and the overall prognosis. However, not all congenital gastrointestinal malformations have a severity score analysis universally accepted (such as Meckel’s diverticulum, duodenal atresia or hypertrophic pyloric stenosis). For the remaining anomalies having a severity scoring (Waterstone classification for esophageal atresia, Krikenbeck classification for anorectal malfromation), data consistent with scoring system for patients included was not reported, thus precluding consistent severity classification and limiting comparison of outcomes.
We have added clarrifying note in the discussion section. (page 12, line 468-471)

Reviewer 2 Report
Comments and Suggestions for Authors
Dear Authors,
This is an interesting and valuable study that systematically reports the correlation between COVID-19 and postoperative outcomes in the surgical treatment of congenital gastrointestinal malformations. The data currently available in the literature regarding COVID-19 and its effects on the management of pediatric patients remain limited. Therefore, future studies with larger sample sizes and longer follow-up periods are required to further investigate the impact of COVID-19 on congenital gastrointestinal malformations. This study represents an initial attempt to extrapolate the effects of COVID-19 on congenital pediatric malformations. Below, I provide a few comments:
- The research was conducted in a precise and accurate manner, and the methodology is explained clearly and comprehensively.
- Line 25: there is a typographical error; the correct name is “Joanna Briggs Institute”.
- The topic is of scientific relevance, and the findings may contribute to the current body of evidence on COVID-19 and surgical outcomes in the pediatric population.
- Please carefully review the manuscript once more in terms of English language, preferably with the assistance of a native speaker, as there are some grammatical and syntactical errors that should be corrected to improve fluency
The quality of English is fine, but I suggest to review the manuscript once again in terms of English language to improve its fluency
Author Response
Reviewer 2 – comments and responses
Comment 1 - This is an interesting and valuable study that systematically reports the correlation between COVID-19 and postoperative outcomes in the surgical treatment of congenital gastrointestinal malformations. The data currently available in the literature regarding COVID-19 and its effects on the management of pediatric patients remain limited. Therefore, future studies with larger sample sizes and longer follow-up periods are required to further investigate the impact of COVID-19 on congenital gastrointestinal malformations. This study represents an initial attempt to extrapolate the effects of COVID-19 on congenital pediatric malformations. Below, I provide a few comments. The research was conducted in a precise and accurate manner, and the methodology is explained clearly and comprehensively.
Response 1 –
We thank the reviewer for the constructive feedback and appreciate the acknowledgment of our study as an initial contribution to understanding the impact of COVID-19 on postoperative outcomes in congenital gastrointestinal malformations.
We agree that the available literature is limited and that further multicenter studies with larger cohorts and longer follow-up are warranted. These points are underscored in Discussion section (line 480 - 486).
Comment 2 - Line 25: there is a typographical error; the correct name is “Joanna Briggs Institute”.
Response 2 –
As suggested, typographical errors highlighted have beed revised accordingly. (Joana -> Joanna).
Comment 3 – The topic is of scientific relevance, and the findings may contribute to the current body of evidence on COVID-19 and surgical outcomes in the pediatric population.
Comment 4 – Please carefully review the manuscript once more in terms of English language, preferably with the assistance of a native speaker, as there are some grammatical and syntactical errors that should be corrected to improve fluency.
Comment 5 – The quality of English is fine, but I suggest to review the manuscript once again in terms of English language to improve its fluency.
Response 3/4/5 – We thank the reviwer for the insightfull feedback. The manuscript was re-edited for clarity and fluency and subsequently reviewed by a native English-speaking colleague with experience in academic writing. All language updates are reflected in the revised version.

Reviewer 3 Report
Comments and Suggestions for Authors
General Comments
- The manuscript is generally well-structured, but sections such as the introduction and discussion are overly detailed and repetitive. These should be shortened and focused on aspects directly relevant to congenital gastrointestinal malformations and COVID-19.
- The topic is timely and understudied, but the novelty and contribution of the review need to be highlighted more explicitly in the introduction and conclusion. Currently, the advancement of knowledge provided by this work is not sufficiently clear.
- The English is comprehensible but requires editing for grammar, sentence structure, and conciseness. For example, phrases such as “the clinical course of these patients is generally asymptomatic or mild and did not appear to alter the intraoperative or postoperative surgical plan” could be expressed more succinctly.
Specific Comments
- Abstract
- The results section is too detailed, listing every anomaly. It should be condensed to emphasize the main findings (sample size, most common malformations, outcomes, and the overall impact of COVID-19).
- The conclusion should be more impactful, stressing the clinical implications and research gaps.
- Introduction
- Some epidemiological details (e.g., global mortality, EUROCAT prevalence) are informative but could be streamlined. The focus should shift to what is known and unknown about COVID-19’s impact on congenital GI anomalies.
- Clearly state the research gap and the study’s specific objectives at the end of this section.
- Methods
- Clarify whether screening, data extraction, and bias assessment were performed independently by multiple reviewers to minimize selection bias.
- Results
- This section is well presented-clear, concise, and appropriately focused.
- Discussion
- The discussion is comprehensive but somewhat repetitive. Consider grouping findings under clear themes (e.g., perioperative outcomes, COVID-19 impact, protective strategies, limitations).
- Provide a more balanced perspective by addressing conflicting evidence (e.g., studies reporting increased perioperative complications vs. those showing no effect).
- Highlight the very small sample size (n = 29) more strongly, as it critically limits the generalizability of conclusions.
- Ensure all abbreviations are defined at first mention (e.g., PSARP).
- Standardize reference formatting, as several entries are incomplete or contain errors.
- Correct typographical errors (e.g., “hipertrophic” → hypertrophic, “posteoprative” → postoperative).
- Conclusion
- The current version largely restates the findings. It should instead emphasize the clinical message (i.e., COVID-19 does not appear to worsen surgical outcomes, though evidence remains limited).
- Suggest clearer future research directions (e.g., multicenter registries, prospective cohort studies, and long-term follow-up).
Thank you for considering these comments.
Sincerely,
Comments on the Quality of English Language
The English is comprehensible but requires editing for grammar, sentence structure, and conciseness. For example, phrases such as “the clinical course of these patients is generally asymptomatic or mild and did not appear to alter the intraoperative or postoperative surgical plan” could be expressed more succinctly.
Author Response
Reviewer 3 – comments and responses
General Comments
Comment 1 - The manuscript is generally well-structured, but sections such as the introduction and discussion are overly detailed and repetitive. These should be shortened and focused on aspects directly relevant to congenital gastrointestinal malformations and COVID-19.
Response 1 –
We are greatfull for reviewer’s thorough assessment. As recommended, we have revised the manuscript introduction and discussion section removing redundant information and focusing more specifically on the relevant aspects between congenital gastrointestinal malformation – SARS-CoV-2 infection.
Comment 2 - The topic is timely and understudied, but the novelty and contribution of the review need to be highlighted more explicitly in the introduction and conclusion. Currently, the advancement of knowledge provided by this work is not sufficiently clear.
Response 2 -
To address this point, we have revised the introduction and conclusion to emphasize more clearly the novelty and scientific contribution of our review. The revisions highlight how this work adds to current knowledge by synthesizing the limited evidence on postoperative outcomes of congenital gastrointestinal malformations in the context of COVID-19 and by identifying existing research gaps. (line 92-107, 486 – 496)
Comment 3 - The English is comprehensible but requires editing for grammar, sentence structure, and conciseness. For example, phrases such as “the clinical course of these patients is generally asymptomatic or mild and did not appear to alter the intraoperative or postoperative surgical plan” could be expressed more succinctly.
Response 3 –
Following this observation, the manuscript was re-edited for clarity and fluency and subsequently reviewed by a native English-speaking colleague with experience in academic writing. All language updates are reflected in the revised version.
Specific Comments
Abstract
Comment 1 – The results section is too detailed, listing every anomaly. It should be condensed to emphasize the main findings (sample size, most common malformations, outcomes, and the overall impact of COVID-19). The conclusion should be more impactful, stressing the clinical implications and research gaps.
Response 1 –
In accordance with the reviewr’s comment, we have revised the result section to focus more on key findings – sample size, type of malformations and overall outcome. Additionaly, the conclusion section has been updated to better reflect clinical implications and highlight the need for further research.
Introduction
Comment 1 – Some epidemiological details (e.g., global mortality, EUROCAT prevalence) are informative but could be streamlined. The focus should shift to what is known and unknown about COVID-19’s impact on congenital GI anomalies. Clearly state the research gap and the study’s specific objectives at the end of this section.
Response 1 –
This observation helped refine the introduction section which has been updated accordingly.
Methods
Comment 1 – Clarify whether screening, data extraction, and bias assessment were performed independently by multiple reviewers to minimize selection bias.
Response 1 –
Following this commnet, we have updated the clarrificatio. Screening, data extraction, and risk-of-bias assessment were each performed independently by two reviewers to minimize selection bias. This clarification has been added to the Methos section – line 560 / 578.
Results
Comment 1 – This section is well presented-clear, concise, and appropriately focused.
Response 1 –
Discussion
Comment 1 – The discussion is comprehensive but somewhat repetitive. Consider grouping findings under clear themes (e.g., perioperative outcomes, COVID-19 impact, protective strategies, limitations). Provide a more balanced perspective by addressing conflicting evidence (e.g., studies reporting increased perioperative complications vs. those showing no effect). Highlight the very small sample size (n = 29) more strongly, as it critically limits the generalizability of conclusions. Ensure all abbreviations are defined at first mention (e.g., PSARP). Standardize reference formatting, as several entries are incomplete or contain errors. Correct typographical errors (e.g., “hipertrophic” → hypertrophic, “posteoprative” → postoperative).
Response 1 –
We appreciate the reviwer’s comprehensive and constructive feedback. The discussion section have been reorganised to be more comprehensive and less redundant. Conflicting evidence from the literature has been incorporated to provide a more balanced interpretation of perioperative outcomes. The limitation related to the small sample size (n = 29) has been emphasized more explicitly. All abbreviations have been defined upon first mention, typographical errors corrected, and reference formatting standardized.
Conclusion
Comment 1 – The current version largely restates the findings. It should instead emphasize the clinical message (i.e., COVID-19 does not appear to worsen surgical outcomes, though evidence remains limited). Suggest clearer future research directions (e.g., multicenter registries, prospective cohort studies, and long-term follow-up).
Response 1 –
We appreciate the reviewr’s insightfull comment. The discussion and conclusion sections have been revised to underscore that COVID-19 does not appear to negatively influence surgical outcomes in pediatric patients with congenital gastrointestinal malformations, while acknowledging the limitations of current evidence. We have also clarified future research directions, highlighting the need for multicenter registries, prospective cohort studies, and long-term follow-up. These revisions are reflected in the updated Conclusion section (Lines 1785 – 1788).

Round 2
Reviewer 1 Report
Comments and Suggestions for Authors
This review investigated whether COVID-19 affects surgical outcomes for congenital gastrointestinal malformations, including hypertrophic pyloric stenosis (L373-L377). Given the limited number of studies meeting the criteria for analysis, this reviewer suggests that the following analyses be included in the results.
The findings stated in the abstract and conclusion must be clearly demonstrated to readers as data within the results section. Table 3 lists the background and outcomes of COVID-19 infection cases by literature source, but it fails to utilize the stratification of surgical invasiveness performed in Table 2. Table 3 can remain as background information. However, please create a new table (a 3 x 2 cross-tabulation table) categorizing the cases in Table 3 into three levels of surgical invasiveness (low, moderate, high). Within each category, present the number and percentage of cases with COVID-19-related complications and the number and percentage of cases with surgery-related complications.
Furthermore, test for independence using a 3 x 2 cross-tabulation table to determine whether there is a significant difference in the presence of COVID-19-related complications based on surgical invasiveness, and whether there is a significant difference in the presence of surgical complications based on the level of surgical invasiveness. If the authors' conclusion is correct, independence should be maintained for surgical invasiveness in COVID-19-related complications, while independence should be rejected for surgical complications.
The following points are listed for revision.
1) Is "congenital gut defects" on L-18 used interchangeably with "congenital gastrointestinal malformations"? The relationship between these two terms is unclear.
2) Do lines L-313 to L-319 refer to the content of Reference 25? The citation number is missing.
Author Response
Responses for reviewers part 2
Reviewer 1 –
Comment 1 –
This review investigated whether COVID-19 affects surgical outcomes for congenital gastrointestinal malformations, including hypertrophic pyloric stenosis (L373-L377). Given the limited number of studies meeting the criteria for analysis, this reviewer suggests that the following analyses be included in the results.
The findings stated in the abstract and conclusion must be clearly demonstrated to readers as data within the results section. Table 3 lists the background and outcomes of COVID-19 infection cases by literature source, but it fails to utilize the stratification of surgical invasiveness performed in Table 2. Table 3 can remain as background information. However, please create a new table (a 3 x 2 cross-tabulation table) categorizing the cases in Table 3 into three levels of surgical invasiveness (low, moderate, high). Within each category, present the number and percentage of cases with COVID-19-related complications and the number and percentage of cases with surgery-related complications.
Response 1 –
We appreciate the revierwer for this constructive reccomendation. In response to the reviewer’s suggestion, we have revised the results. As table 3 was complex already, table 2 was modified to include data on surgical invasivenes derived from the reported operative approach and the presence of an anastomosis. This modification enables consistent stratification of procedures into low, moderate, and high invasiveness categories that were subsequently applied for the comparative analysis presented in the newly created 3 × 2 cross-tabulation table. (table 3). Results section were amended accordingly (line 304 – 312)
Surgical invasiveness was classified based on operative approach and extent of bowel manipulation as follows –
Low – minimally invasive procedures without bowel lumen opening or anastomosis – congenital hypertrophic pyloric stenosis.
Moderate – procedures involving gastrointestinal anastomosis via minimally invasive approach or transanally –
High – Open/ Minimally invasive converted to open surgical approach with bowel anastomosis or other major reconstructive procedures –
Comment 2 – Furthermore, test for independence using a 3 x 2 cross-tabulation table to determine whether there is a significant difference in the presence of COVID-19-related complications based on surgical invasiveness, and whether there is a significant difference in the presence of surgical complications based on the level of surgical invasiveness. If the authors' conclusion is correct, independence should be maintained for surgical invasiveness in COVID-19-related complications, while independence should be rejected for surgical complications.
Response 2 –
We appreciate the reviewer’s detailed statistical guidance. Because the total sample was small (n = 13) and several cells contained zero expected counts, we applied the Fisher–Freeman–Halton exact test. The exact analyses indicated no statistically significant association between the level of surgical invasiveness and the presence of COVID-19–related complications (p = 0.30) or surgery-related complications (p = 0.47). Although all observed complications occurred within the high-invasiveness group, the small sample size limits statistical power. These findings have been incorporated into the Methods, Results and Discussion sections of the revised manuscript. (line 194 – 198, 342 - 346, 442-452)
The following points are listed for revision.
Comment 4 – Is "congenital gut defects" on L-18 used interchangeably with "congenital gastrointestinal malformations"? The relationship between these two terms is unclear.
Response 4 –
We thank the reviewer for this valuable observation. The term “congenital gut defects” was intended to refer to structural gastrointestinal anomalies of congenital origin. However, to ensure the consistency with established pediatric surgical literature and to improve precision, we have replaced the phrase “congenital gut defects” with congenital gastrointestinal malformation.
The study analyzed the incidence and distribution of congenital gastrointestinal malformations in neonates. – line 18
Comment 5 – Do lines L-313 to L-319 refer to the content of Reference 25? The citation number is missing.
Response 5 –
We appreciate the reviwer’s attention to detail. The text lines L-313 to L-319 indeed refers to the study cited as Reference number 25. As recommended, we have corrected this omission and added the appropriate citation number at the end of the paragraph to maintan the accuracy of data reported.

Reviewer 3 Report
Comments and Suggestions for Authors
No
Author Response
NA.